# Physical compatibility of Xuebijing injection with 53 intravenous drugs during simulated Y-site administration

Tong Tong[1,2☯], Peifang Li[1,2☯], Haiwen Ding[1,2], Ying Huang[1,2], Sheng Liu[1,2]*

1 Department of Pharmacy, The First Affiliated Hospital of USTC, Division of Life Sciences and Medicine, University of Science and Technology of China, Hefei, Anhui, P.R. China, 2 Anhui Provincial Key Laboratory of Precision Pharmaceutical Preparations and Clinical Pharmacy, Hefei, Anhui, China

☯ These authors contributed equally to this work.
* lslcclhl@ustc.edu.com.cn

**Data Availability Statement:** All relevant data are within the manuscript and its Supporting Information files.

**Funding:** The author(s) received no specific funding for this work.

## Abstract

### Objective

Xuebijing injection (XBJ) is a commonly used herbal medicine injection in China. However, the physical compatibility of XBJ with other intravenous drugs remains unclear. The purpose of this research is to evaluate physical compatibility of Xuebijing injection (XBJ) with 53 intravenous drugs (including 31 Chinese medicine injections and 22 chemicals) during simulated Y-site administration.

### Methods

Y-site administration was simulated in vitro by admixing 0.33 ml/ml XBJ with an equal volume of other diluted 53 intravenous drugs, respectively. Physical compatibility including visual inspection, Tyndall beam, particle limits, turbidity, pH, chromacity value, spectroscopic absorption of 550 nm and 420 nm ($A_{550\ nm}$ and $A_{420\ nm}$) were observed and assessed at 0, 1, 2, and 4 h. Physical compatibility was defined as all solutions with no color changes, no gas evolution, particulate formation and no Tyndall beam within 4 hours, turbidity changes <0.5 nephelometric turbidity unit (NTU) compared to 0 h, particle limits allowed by the Chinese Pharmacopoeia (Ch.P) 2020 edition, pH changes <10% compared to 0, chromacity value changes <200 compared to 0 h, or photometrical changes of $A_{420\ nm}$ <0.0400 or $A_{550\ nm}$ <0.0100 compared to 0 h.

### Results

XBJ was physically incompatible with 27 of the 53 intravenous drugs tested, 26 were compatible with XBJ for 4 h.

### Conclusions

XBJ should not be simultaneously co-administered with 27 of the 53 intravenous drugs during simulated Y-site. If coadministration was inevitable, flushing tube with NS or D5W before and after infusion of XBJ was needed. Assessment included visual inspection, Tyndall

**Competing interests:** The authors have declared that no competing interests exist.

beam, turbidity measurement, particle counts, pH measurement, chromacity value measurement and absorption of $A_{550\,nm}$ were proved to be valid and robust for the quality control of infusion and compatibility of Chinese herbal injection.

## 1. Introduction

Xuebijing injection (XBJ) is a yellow or brownish-yellow clarified liquid mainly comprising of Honghua (Flos Carthami), Chishao (Radix Paeoniae Rubra), Chuanxiong (Rhizoma Chuanxiong), Danshen (Radix Salviae Miltiorrhiae) and Danggui (RadixAngelicae Sinensis) [1], the excipients are glucose, polysorbate 80 and water for injection. It was licensed in 2004 by the National Medical Products Administration (NMPA, China) [2] and approved for warm and heat diseases, such as fever, shortness of breath, palpitation, irritability and other syndrome of stasis and poison, to treat patients suffering from systemic inflammatory response syndrome induced by infection, combining therapy with the treatment of multiple organ dysfunction syndrome in the period of organ dysfunction. It can also treat severe and critical systemic inflammatory response syndrome and/or multiple organ failure of novel coronavirus infection.

As a commonly used herbal medicine injection in China, XBJ has been administered to critically ill patients for over 15 years. Extensive studies and clinical research have revealed its functions in sepsis [2–4]. The injection exhibits protective mechanisms by antagonizing endotoxins and inhibiting the uncontrolled release of inflammatory mediators from endotoxin-stimulated monocytes/macrophages. Additionally, XBJ improves coagulopathy in disseminated intravascular coagulation, an important risk factor for sepsis mortality [5]. In China, over 250,000 patients receive XBJ treatment annually [6], and its route of administration and safety are well-established. The multiple positive clinical effects of XBJ make it an essential drug for critically ill patients. However, managing the coadministration of multiple drugs, especially those requiring prolonged intravenous infusions, along with XBJ can present challenges.

Due to limited independent catheters for central venous catheters [7], infusions should be ceased and the line flushed before administering other drugs [8]. However, frequent stopping and flushing may affect the patient's fluid balance [9]. Y-site access enables simultaneous intravenous drug coadministration but can lead to physical incompatibilities between drugs. Our main interest lies in the physical compatibility of XBJ with other intravenous drugs through Y-site. The quality and safety of co-administering XBJ with intravenous drugs remain unclear. This study investigates the physical compatibility of 53 commonly used intravenous drugs (31 Chinese medicine injections and 22 chemicals) with XBJ during Y-site administration. Methods used include visual inspection, Tyndall beam, particle limits, turbidity and pH changes, chromacity value changes, and spectroscopic absorption at 550 nm and 420 nm over 4 hours. Determining physical compatibility between XBJ and other intravenous drugs via Y-site could enhance safety management, reduce nursing time, and improve clinical utility. Additionally, we evaluated the sensitivity of these methods for scientific and feasible compatibility testing of Chinese herbal injections with other intravenous drugs to enhance infusion quality control.

## 2. Materials and methods

### 2.1 Sample preparation

Under aseptic and laminar air flow conditions, the selected 53 intravenous drugs were diluted in NS or D5W as recommended by the manufacturer, a total of 50 ml XBJ was slowly diluted

in either NS or D5W to a concentration of 0.33 ml/ml. Selected drug solutions were slowly added into the diluted solution of XBJ, respectively. Each of sample solutions was passed through a 0.5 μm filter, XBJ solution and selected drug solutions were gradually combined during a single test, followed by mixing the resulting solutions with a plugged glass tube or an empty venous infusion bag at a 1:1 ratio. The final concentration of the mixed solutions was half that of a single drug.

All solutions were gently inverted three times at room temperature (approximately 22˚C) at 0, 1, 2, and 4 h to ensure complete mixing. The solutions were then allowed to stand briefly or subjected to ultrasonic defoaming for 1 min. Physical compatibility including visual inspection, Tyndall beam, turbidity, particle counts, pH, chromacity value, $A_{550\ nm}$ and $A_{420\ nm}$ were observed and assessed intervals 0, 1, 2, and 4 h. Drug particulars, manufacturer, specification, lot number, diluent and concentrations were recorded for each drug (Table 1).

## 2.2 Experimental controls

2 negative control solutions containing 0.33 ml/ml XBJ in D5W and 0.33 ml/ml XBJ in NS, 4 positive control solutions containing 2.5 mg/ml calcium chloride with 0.0025 ml/ml composite potassium in NS, 10 μm latex particles reference material and 25 μm particle count reference material (Haianhongmeng reference material technology Co., Ltd, Beijing, Lot: 20221003 and L693, respectively) were used as control solutions to ensure the test results in experiments.

## 2.3 Visual inspection

According to Part 4 of the Chinese Pharmacopoeia (Ch.P) 2020 edition, using the "Solution Color Test Method" and "Clarification Test Method" [10], at each test point, all solutions were examined using a clarity tester on both black and white backgrounds with the naked eye. Visual incompatibility was defined as color changes, gas evolution, or visible particulate formation within 4 hours.

## 2.4 Tyndall beam assessment

Tyndall beam was assessed using a red laser pointer (650 nm, 50 mW) on both black and white backgrounds. Any sample prevented the light from passing through the solutions and appearing Tyndall beam was considered incompatible [11, 12].

## 2.5 Turbidity measurement

Turbidity was measured at each time point through laboratory-grade turbidimeter (INESA Physico optical instrument Co., Ltd, Shanghai) according to the instructions. Three repeated measurements were taken for each sample, using the average as the final result. Physical incompatibility was defined as turbidity changes ≥0.5 NTU compared to 0 h [13, 14].

## 2.6 Particles measurement

Particle counts were counted using the GWF-8JA Particle Counter (Tianjin Tianhe Analysis Instrument Co., Ltd, Tianjin). Part 4 of the Ch.P recommended that injectable solutions be analyzed using a light obscuration particle count test. According to the Ch.P, for injectable solutions labeled as ≥100 ml, the number of ≥10 μm particles shall not exceed 25 particles/ml, the number of ≥25 μm particles shall not exceed 3 particles/ml [10]. Three repeated measurements were taken for each sample, using the average as the final result.

**Table 1. Details of tested drugs in the study.**

| | Drug | Manufacturer | Specification | Lot | Diluent | Drug Concentration (/ml) |
|---|---|---|---|---|---|---|
| 1 | Aciclovir Sodium | Furen medicines group | 0.25 g | 2210237 | NS | 5mg |
| 2 | Ambroxol Hydrochloride | Huazhong Pharmaceutical | 15mg/2ml | F221222A | NS | 0.300 mg |
| 3 | Calcium Chloride | Sichuan Meida Kangjiale Pharmaceutical | 0.5g/10ml | 22062016 | NS | 5.000 mg |
| 4 | Calcium Gluconate | Sichuan Meida Kangjiale Pharmaceutical | 1g/10ml | 22121526 | D5W | 10.000 mg |
| 5 | Cefazolin Sodium | Zhejiang CR Sanjiu Zhongyi Pharmaceutical | 0.5g | JX2204081 | NS | 10.000 mg |
| 6 | Cefoperazone Sodium and Sulbactam Sodium (11) | Livzon Pharmaceutical Group | 2g | DCK721201 | NS | 20.000 mg |
| 7 | Ceftriaxone Sodium | Hunan Kelun Pharmaceutical | 1g | E2211014 | NS | 20.000 mg |
| 8 | Cefuroxime Sodium | Guangzhou Baiyunshan Tianxin Pharmaceutical | 1.5g | 2302384 | NS | 15.000 mg |
| 9 | Ciwujia injection | Duoduo Pharmaceutical | 20ml | 22061012 | NS | 1.000 mg |
| 10 | Composite Potassium Hydrogen Phosphate | Tianjin Jinyao Pharmaceutical | 2ml | 2209131 | NS | 0.005 ml |
| 11 | Danhong injection | Shandong Danhong Pharmaceutical | 20ml | 22082014 | NS | 0.140 ml |
| 12 | Danshen Chuanxiongqin injection | Jilin Sichang Pharmaceutical | 5ml | 20220709 | NS | 0.057 ml |
| 13 | Dazhu Hongjingtian injection | Tonghua Yusheng Pharmaceutical | 5ml | 1001220102 | D5W | 0.038 ml |
| 14 | Dengzhanxixin injection | Yunnan Biovalley Pharmaceutical | 10ml | 20220545 | NS | 0.074 ml |
| 15 | Dexamethasone Sodium Phosphate | Cisen Pharmaceutical | 5mg/1ml | 2205260611 | NS | 0.100 mg |
| 16 | Doxofylline | Zhejiang Beisheng Pharma Hansheng Pharmaceutical | 0.3g/20ml | 2101141 | NS | 3.000 mg |
| 17 | Extract of Ginkgo Biloba Leaves | Youcare Pharmaceutical Group | 17.5mg/5ml | 19821028 | NS | 0.350 mg |
| 18 | Ganciclovir sodium | Wuhan Pusheng Pharmaceutical | 0.25g | 230206–1 | NS | 5.000 mg |
| 19 | Ginkgo leaf Extract and Dipyridamole | Shanxi Pude Pharmaceutical | 10ml | 8220205 | NS | 0.048 ml |
| 20 | Guanxinning injection | Shineway Pharmaceutical | 10ml | 220320B3 | NS | 0.107 ml |
| 21 | Honghua injection | Shineway Pharmaceutical | 10ml | 220126D2 | NS | 0.074 ml |
| 22 | Huangqi injection | Shineway Pharmaceutical | 10ml | 210609A3 | NS | 0.074 ml |
| 23 | HuangqiDuotang for Injection | Tianjin Cinorch Pharmaceutical | 250mg | 220409 | NS | 1.000 mg |
| 24 | Hydrocortisone SodiumSuccinate | Yantai DongchengbeifangPharmaceutica | 50mg | 202208141 | NS | 1.000 mg |
| 25 | Ilaprazole Sodium | Livzon Pharmaceutical Group | 10mg | 210709 | NS | 0.100 mg |
| 26 | Kang'ai injection | Changbaishan Pharmaceutical | 20ml | 2220508 | NS | 0.140 ml |
| 27 | Kuhuang injection | Changshu Lei Yunshang Pharmaceutical | 10ml | 2206211 | NS | 0.108 ml |
| 28 | Lansoprazole Sodium | Aosaikang Pharmaceutical | 30mg | J2201031 | NS | 0.300 mg |
| 29 | Magnesium Isoglycyrrhizinate | Chiatai Tianqing Pharmaceutical | 50mg/10ml | 221030204 | NS | 1.500 mg |
| 30 | Magnesium Sulfate | Shanghai Jindi Jiuzhou Pharmaceutical | 2.5g/10ml | 221041001C | D5W | 25.000 mg |
| 31 | Mailuoning injection | Jinling Pharmaceutical | 10ml | 20220502 | NS | 0.074 ml |
| 32 | Methylprednisolone Sodium Succibate | Huapont Pharmaceutical | 0.5g | 230021212 | NS | 1.000 mg |
| 33 | OmeprazoleSodium | Huabei Pharmaceutical | 40mg | 2AFND20816 | NS | 0.400 mg |
| 34 | Pantoprazole Sodium | Yangtze River Pharmaceutical | 40mg | 22110141 | NS | 0.400 mg |
| 35 | Potassium Chloride | China Otsuka Pharmaceutical | 1.5g/10ml | 2G92K2 | NS | 3.000 mg |
| 36 | Qingkailing injection | Shineway Pharmaceutical | 10ml | 220126D2 | NS | 0.074 ml |
| 37 | Rabeprazole Sodium | Aosaikang Pharmaceutical | 20mg | J2208081 | NS | 0.200 mg |
| 38 | Reduced Glutathione | YaoPharma | 0.9g | 22232450 | NS | 18.000 mg |
| 39 | Reduning injection | Jiangsu Kanion Pharmaceutical | 10ml | 220506 | NS | 0.074 ml |
| 40 | Shenfu injection | CR Sanjiu Ya'an Pharmaceutical | 10ml | 211101AK04 | D5W | 0.074 ml |
| 41 | Shenmai injection | Chiatai Qingchunbao Pharmaceutical | 50ml | 2202227 | - | - |
| 42 | Shenqi Fuzheng injection | Livzon Group Limin Pharmaceutical | 250ml | 220537 | - | - |
| 43 | Shuanghuanglian injection | Henan Fusen Pharmaceutical | 20ml | 2203211 | NS | 0.029 ml |
| 44 | Shuxuetong injection | Mudanjiang Youbo Pharmaceutical | 2ml | 22011202 | NS | 0.024 ml |

(*Continued*)

**Table 1.** (Continued)

| | Drug | Manufacturer | Specification | Lot | Diluent | Drug Concentration (/ml) |
|---|---|---|---|---|---|---|
| 45 | Sulfotanshinone sodium | Shanghai Shangyao Diyishenghua Pharmaceutical | 10mg/2ml | 2303107 | NS | 0.160 mg |
| 46 | Tanreqing injection | Shanghai Kaibao Pharmaceutical | 10ml | 2206301 | NS | 0.074 ml |
| 47 | Xiangdan injection | Chiatai Qingchunbao Pharmaceutical | 10ml | 2203043 | NS | 0.074 ml |
| 48 | Xingnaojing injection | Wuxi Jiyushanhe Pharmaceutica | 10ml | 210807 | NS | 0.057 ml |
| 49 | Xinmailong injection | Yunnan Tengyao Pharmaceutica | 2ml | 2205101 | NS | 1.600 mg |
| 50 | Yinxingneizhi injection | Chengdu Baiyu Pharmaceutical | 2ml | 022110073 | NS | 0.200 mg |
| 51 | Xiyanping injection | Jiangxi Qinfeng Pharmaceutical | 50mg/2ml | 220714 | NS | 1.000 mg |
| 52 | Xuesaitong injection | Lonch GroupWanrong Pharmaceutical | 20ml | 22050211 | D5W | 1.600 mg |
| 53 | Xueshuantong for injection | Guangxi Wuzhou Pharmaceutical | 150mg | 22010321 | NS | 1.800 mg |
| 54 | XBJ | Tianjin Hongri Pharmaceutical | 10ml | 2302061 | D5W | 0.33ml |
| 55 | XBJ | Tianjin Hongri Pharmaceutical | 10ml | 2302061 | NS | 0.33ml |
| 56 | NS | Fengyuan Pharmaceutical | 100ml | 3122090602 | - | - |
| 57 | D5W | Fengyuan Pharmaceutical | 100ml | 3122080801 | - | - |

## 2.7 pH measurement

pH was measured to assess whether acid-base reactions may involve any observed incompatibilities. The same time points mentioned above (intervals 0, 1, 2, and 4 h), pH of solution were determined using a pH meter (Qiwei instrument Co., Ltd, Hangzhou). Any solutions with pH variations <10% compared to the baseline (immediately after mixing) were considered incompatible [15].

## 2.8 Chromacity value measurement

According to the Ch.P, "Solution Color Test Method" described using a colorimeter (Qiwei instrument Co., Ltd, Hangzhou) to determine chromacity value of the solutions [10]. It was stipulated to use yellow tonal standard solutions (Haianhongmeng reference material technology Co., Ltd, Beijing, Lot: M751) for comparison which contain 0.5, 1, 2, 3, 4, 5, 6, 7, 8, 9, 10 color codes. The chromacity value of 0.5 to 10 were about 25, 50, 100, 150, 200, 250, 300, 400, 500, 600, 700, respectively. If the chromacity value changes >200, and the color of the solution changes visually, solution color was defined as physical incompatibility.

## 2.9 Spectroscopic measurement

The Spectroscopic measurement was performed using 8453 Hewlett Packard diode array ultraviolet visible spectrophotometer to detect any indications of color change ($A_{420\ nm}$) or haze ($A_{550\ nm}$). All solutions were considered incompatible if the absorption changes by $A_{420\ nm}$ >0.0400 or $A_{550\ nm}$ >0.0100 [16].

## 2.10 Statistical analysis

Descriptive statistics and original data were presented. Results of particle count were reported as mean ± standard deviations (mean ± SD). No further statistical analysis was performed.

## 2.11 Definition of compatibility

Justification of compatibility refers to the diagram in Fig 1. Physical compatibility was defined as all solutions with no color changes, no gas evolution, particulate formation and no Tyndall

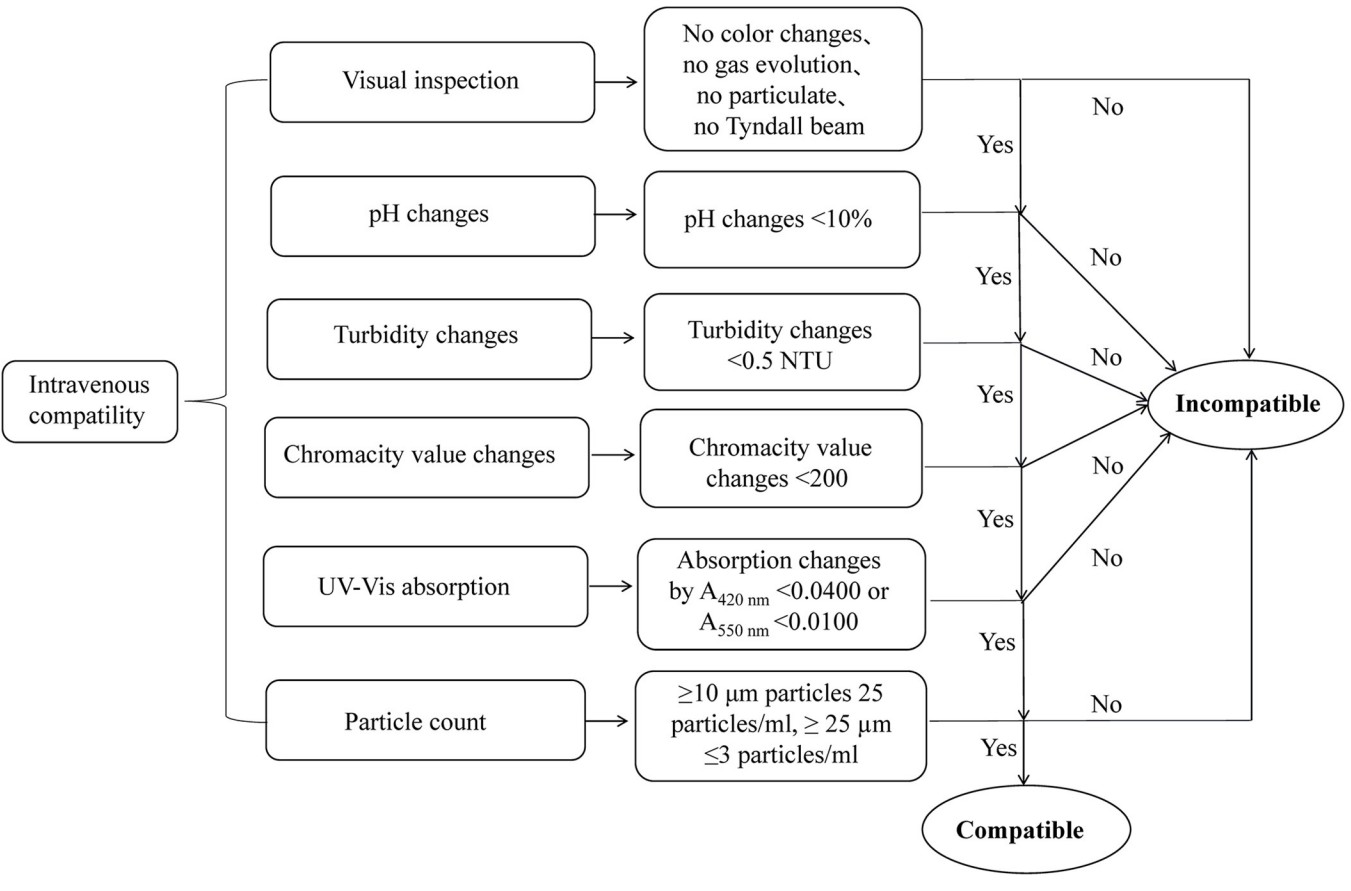

**Fig 1. Diagram of physical compatibility justification.**

beam within 4 hours, turbidity changes <0.5 NTU compared to 0 h, particle limits allowed over the Ch.P, pH changes <10% compared to 0, chromacity value changes <200 compared to 0 h, or absorption changes by $A_{420 \, nm}$ >0.0400 or $A_{550 \, nm}$ >0.0100 compared to 0 h.

## 3 Results

### 3.1 Visual inspection and Tyndall beam findings

No visual changes (color changes, gas evolution, haze, or visible particulate formation) were detected over the 4 hours period for the solutions (Table 2). However, the combination of XBJ + Acyclovir sodium, XBJ + Ceftriaxone sodium, XBJ + Cefuroxime sodium, XBJ + Iprazole sodium and XBJ + Rabeprazole sodium solutions in NS displayed Tyndall Beam, respectively (Fig 2), indicating that XBJ was incompatible with above 5 drugs.

### 3.2 Turbidity changes findings

The results from turbidity measurement were summarized in Table 2. Of the 53 selected drugs, combinations of XBJ + 51 drugs result in turbidity changes <0.5 NTU compared to 0 h, respectively. At the time of 1, 2, 4 h, combinations of XBJ + Iprazole sodium and XBJ + Rabe-prazole sodium solutions displayed turbidity changes >0.5 NTU compared to 0 h. It indicated that XBJ was incompatible with Iprazole sodium and Rabeprazole sodium.

**Table 2. Findings of visual inspection, Tyndall beam, turbidity changes in solutions.**

| Drug | Color/Clarity (White background) | | | | Tyndall beam (Black background) | | | | Turbidity and the changes (NTU) | | | | Compatibility |
|---|---|---|---|---|---|---|---|---|---|---|---|---|---|
| | 0 h | 1 h | 2 h | 4 h | 0 h | 1 h | 2 h | 4 h | 0 h | 1 h | 2 h | 4 h | |
| Aciclovir Sodium | Brown/Clear | Brown/Clear | Brown/Clear | Brown/Clear | P | P | P | P | 0.202 | 0.047 | -0.002 | 0.042 | Incomp |
| Ambroxol Hydrochloride | Yellow/Clear | Yellow/Clear | Yellow/Clear | Yellow/Clear | N | N | N | N | 0.061 | 0.031 | 0.025 | 0.021 | Comp |
| Calcium Chloride | Yellow/Clear | Yellow/Clear | Yellow/Clear | Yellow/Clear | N | N | N | N | 0.106 | -0.049 | 0.026 | -0.046 | Comp |
| Calcium Gluconate | Yellow/Clear | Yellow/Clear | Yellow/Clear | Yellow/Clear | N | N | N | N | 0.064 | 0.017 | -0.014 | 0.013 | Comp |
| Cefazolin Sodium | Yellow/Clear | Yellow/Clear | Yellow/Clear | Yellow/Clear | N | N | N | N | 0.156 | 0.014 | -0.022 | -0.024 | Comp |
| Cefoperazone Sodium and Sulbactam Sodium (11) | Yellow/Clear | Yellow/Clear | Yellow/Clear | Yellow/Clear | N | N | N | N | 0.096 | -0.002 | 0.022 | 0.073 | Comp |
| Ceftriaxone Sodium | Yellow/Clear | Yellow/Clear | Yellow/Clear | Yellow/Clear | P | P | P | P | 0.135 | 0.145 | -0.015 | -0.007 | Incomp |
| Cefuroxime Sodium | Yellow/Clear | Yellow/Clear | Yellow/Clear | Yellow/Clear | N | N | P | P | 0.134 | 0.025 | 0.088 | -0.007 | Incomp |
| Ciwujia injection | Yellow/Clear | Yellow/Clear | Yellow/Clear | Yellow/Clear | N | N | N | N | 0.106 | -0.012 | -0.014 | -0.032 | Comp |
| Composite Potassium Hydrogen Phosphate | Yellow/Clear | Yellow/Clear | Yellow/Clear | Yellow/Clear | N | N | N | N | 0.056 | 0.035 | 0.022 | 0.019 | Comp |
| Danhong injection | Yellow/Clear | Yellow/Clear | Yellow/Clear | Yellow/Clear | N | N | N | N | 0.130 | -0.042 | -0.038 | -0.046 | Comp |
| Danshen Chuanxiongqin injection | Yellow/Clear | Yellow/Clear | Yellow/Clear | Yellow/Clear | N | N | N | N | 0.111 | 0.003 | 0.016 | -0.009 | Comp |
| Dazhu Hongjingtian injection | Yellow/Clear | Yellow/Clear | Yellow/Clear | Yellow/Clear | N | N | N | N | 0.066 | -0.018 | 0.050 | 0.045 | Comp |
| Dengzhanxixin injection | Yellow/Clear | Yellow/Clear | Yellow/Clear | Yellow/Clear | N | N | N | N | 0.083 | 0.007 | 0.033 | 0.042 | Comp |
| Dexamethasone Sodium Phosphate | Yellow/Clear | Yellow/Clear | Yellow/Clear | Yellow/Clear | N | N | N | N | 0.109 | 0.029 | 0.056 | 0.000 | Comp |
| Doxofylline | Yellow/Clear | Yellow/Clear | Yellow/Clear | Yellow/Clear | N | N | N | N | 0.040 | 0.058 | 0.002 | 0.021 | Comp |
| Extract of Ginkgo Biloba Leaves | Yellow/Clear | Yellow/Clear | Yellow/Clear | Yellow/Clear | N | N | N | N | 0.122 | -0.026 | -0.006 | -0.058 | Comp |
| Ganciclovir sodium | Yellow/Clear | Yellow/Clear | Yellow/Clear | Yellow/Clear | N | N | N | N | 0.131 | -0.027 | 0.013 | -0.009 | Comp |
| Ginkgo leaf Extract and Dipyridamole | Yellow/Clear | Yellow/Clear | Yellow/Clear | Yellow/Clear | N | N | N | N | 0.062 | 0.010 | 0.003 | 0.025 | Comp |
| Guanxinning injection | Yellow/Clear | Yellow/Clear | Yellow/Clear | Yellow/Clear | N | N | N | N | 0.145 | -0.026 | -0.048 | -0.025 | Comp |
| Honghua injection | Yellow/Clear | Yellow/Clear | Yellow/Clear | Yellow/Clear | N | N | N | N | 0.066 | 0.030 | 0.067 | -0.015 | Comp |
| Huangqi injection | Yellow/Clear | Yellow/Clear | Yellow/Clear | Yellow/Clear | N | N | N | N | 0.072 | 0.013 | -0.007 | 0.008 | Comp |
| HuangqiDuotang for Injection | Yellow/Clear | Yellow/Clear | Yellow/Clear | Yellow/Clear | N | N | N | N | 0.219 | 0.041 | 0.088 | 0.065 | Comp |
| Hydrocortisone SodiumSuccinate | Yellow/Clear | Yellow/Clear | Yellow/Clear | Yellow/Clear | N | N | N | N | 0.105 | -0.040 | -0.049 | -0.047 | Comp |
| Ilaprazole Sodium | Yellow/Clear | Yellow/Clear | Yellow/Clear | Yellow/Clear | P | P | P | P | 0.295 | 1.362 | 1.755 | 2.422 | Incomp |

*(Continued)*

**Table 2.** (Continued)

| Drug | Color/Clarity (White background) | | | | Tyndall beam (Black background) | | | | Turbidity and the changes (NTU) | | | | Compatibility |
|---|---|---|---|---|---|---|---|---|---|---|---|---|---|
| | 0 h | 1 h | 2 h | 4 h | 0 h | 1 h | 2 h | 4 h | 0 h | 1 h | 2 h | 4 h | |
| Kang'ai injection | Yellow/ Clear | Yellow/Clear | Yellow/Clear | Yellow/Clear | N | N | N | N | 0.122 | 0.026 | -0.059 | -0.004 | Comp |
| Kuhuang injection | Yellow/ Clear | Yellow/Clear | Yellow/Clear | Yellow/Clear | N | N | N | N | 0.159 | -0.003 | -0.023 | -0.024 | Comp |
| Lansoprazole Sodium | Yellow/ Clear | Yellow/Clear | Yellow/Clear | Yellow/Clear | N | N | N | N | 0.070 | 0.025 | 0.009 | 0.029 | Comp |
| Magnesium Isoglycyrrhizinate | Yellow/ Clear | Yellow/Clear | Yellow/Clear | Yellow/Clear | N | N | N | N | 0.095 | -0.041 | -0.006 | 0.020 | Comp |
| Magnesium Sulfate | Yellow/ Clear | Yellow/Clear | Yellow/Clear | Yellow/Clear | N | N | N | N | 0.075 | 0.046 | -0.008 | 0.036 | Comp |
| Mailuoning injection | Yellow/ Clear | Yellow/Clear | Yellow/Clear | Yellow/Clear | N | N | N | N | 0.105 | 0.008 | 0.005 | 0.001 | Comp |
| Methylprednisolone Sodium Succibate | Yellow/ Clear | Yellow/Clear | Yellow/Clear | Yellow/Clear | N | N | N | N | 0.093 | -0.016 | -0.032 | -0.023 | Comp |
| OmeprazoleSodium | Yellow/ Clear | Yellow/Clear | Yellow/Clear | Yellow/Clear | N | N | N | N | 0.063 | -0.002 | -0.007 | 0.000 | Comp |
| Pantoprazole Sodium | Yellow/ Clear | Yellow/Clear | Yellow/Clear | Yellow/Clear | N | N | N | N | 0.058 | 0.012 | 0.021 | 0.013 | Comp |
| Potassium Chloride | Yellow/ Clear | Yellow/Clear | Yellow/Clear | Yellow/Clear | N | N | N | N | 0.037 | 0.049 | 0.036 | 0.073 | Comp |
| Qingkailing injection | Yellow/ Clear | Yellow/Clear | Yellow/Clear | Yellow/Clear | N | N | N | N | 0.088 | -0.008 | -0.039 | -0.005 | Comp |
| Rabeprazole Sodium | Yellow/ Clear | Yellow/Clear | Yellow/Clear | Yellow/Clear | P | P | P | P | 0.222 | 1.533 | 4.270 | 6.773 | Incomp |
| Reduced Glutathione | Yellow/ Clear | Yellow/Clear | Yellow/Clear | Yellow/Clear | N | N | N | N | 0.197 | -0.003 | -0.004 | 0.002 | Comp |
| Reduning injection | Yellow/ Clear | Yellow/Clear | Yellow/Clear | Yellow/Clear | N | N | N | N | 0.062 | 0.009 | -0.002 | 0.014 | Comp |
| Shenfu injection | Yellow/ Clear | Yellow/Clear | Yellow/Clear | Yellow/Clear | N | N | N | N | 0.197 | 0.112 | 0.007 | 0.071 | Comp |
| Shenmai injection | Yellow/ Clear | Yellow/Clear | Yellow/Clear | Yellow/Clear | N | N | N | N | 0.314 | 0.037 | -0.040 | -0.075 | Comp |
| Shenqi Fuzheng injection | Yellow/ Clear | Yellow/Clear | Yellow/Clear | Yellow/Clear | N | N | N | N | 0.067 | 0.011 | -0.006 | 0.002 | Comp |
| Shuanghuanglian injection | Yellow/ Clear | Yellow/Clear | Yellow/Clear | Yellow/Clear | N | N | N | N | 0.093 | -0.005 | -0.014 | -0.001 | Comp |
| Shuxuetong injection | Yellow/ Clear | Yellow/Clear | Yellow/Clear | Yellow/Clear | N | N | N | N | 0.074 | -0.021 | 0.040 | 0.130 | Comp |
| Sulfotanshinone sodium | Red/Clear | Red/Clear | Red/Clear | Red/Clear | N | N | N | N | 0.122 | -0.013 | -0.043 | -0.022 | Comp |
| Tanreqing injection | Yellow/ Clear | Yellow/Clear | Yellow/Clear | Yellow/Clear | N | N | N | N | 0.055 | 0.022 | -0.015 | 0.084 | Comp |
| Xiangdan injection | Yellow/ Clear | Yellow/Clear | Yellow/Clear | Yellow/Clear | N | N | N | N | 0.113 | 0.023 | 0.012 | 0.011 | Comp |
| Xingnaojing injection | Yellow/ Clear | Yellow/Clear | Yellow/Clear | Yellow/Clear | N | N | N | N | 0.089 | -0.010 | -0.011 | -0.012 | Comp |
| Xinmailong injection | Yellow/ Clear | Yellow/Clear | Yellow/Clear | Yellow/Clear | N | N | N | N | 0.085 | -0.017 | -0.003 | 0.055 | Comp |
| Yinxingneizhi injection | Yellow/ Clear | Yellow/Clear | Yellow/Clear | Yellow/Clear | N | N | N | N | 0.080 | 0.019 | 0.018 | 0.032 | Comp |

(*Continued*)

**Table 2.** (Continued)

| Drug | Color/Clarity (White background) | | | | Tyndall beam (Black background) | | | | Turbidity and the changes (NTU) | | | | Compatibility |
|---|---|---|---|---|---|---|---|---|---|---|---|---|---|
| | 0 h | 1 h | 2 h | 4 h | 0 h | 1 h | 2 h | 4 h | 0 h | 1 h | 2 h | 4 h | |
| Xiyanping injection | Yellow/Clear | Yellow/Clear | Yellow/Clear | Yellow/Clear | N | N | N | N | 0.100 | 0.013 | -0.045 | -0.036 | Comp |
| Xuesaitong injection | Yellow/Clear | Yellow/Clear | Yellow/Clear | Yellow/Clear | N | N | N | N | 0.084 | 0.014 | -0.022 | -0.036 | Comp |
| Xueshuantong for injection | Yellow/Clear | Yellow/Clear | Yellow/Clear | Yellow/Clear | N | N | N | N | 0.140 | -0.064 | -0.029 | -0.027 | Comp |
| XBJ[A] | Yellow/Clear | Yellow/Clear | Yellow/Clear | Yellow/Clear | N | N | N | N | 0.102 | 0.008 | -0.0013 | -0.003 | Comp |
| XBJ[B] | Yellow/Clear | Yellow/Clear | Yellow/Clear | Yellow/Clear | N | N | N | N | 0.112 | 0.024 | 0.051 | 0.028 | Comp |
| Calcium Chloride with Composite Potassium Hydrogen Phosphate[C] | White/Turbid | White Precipitate | White Precipitate | White Precipitate | P | P | P | P | 3.405 | 67.825 | - | - | Incomp |
| 10 µm latex particles reference material[D] | Colourless/Clear | | | | P | | | | - | - | - | - | |
| 25 µm particle count reference material[E] | Colourless/Clear | | | | P | | | | - | - | - | - | |

**Note:** (A) Negative control, XBJ (Xuebijing injection) in D5W (dextrose 5% water). (B) Negative control, XBJ in NS (Normal saline). (C) Positive control, Calcium Chloride with Composite Potassium in NS. (D) Positive control, 10 µm latex particles reference material. (E) Positive control, 25 µm particle count reference material, P; Tyndall Positive, N; Tyndall Negative, Comp; compatible, Incomp; incompatible

### 3.3 Particle count findings

Table 3 showed the results of the particle count findings. After mixing, a massive growth in number of particles in group of XBJ with Danshen Chuanxiongqin injection, Dengzhanxixin injection, Guanxinning injection, Kang'ai injection, Kuhuang injection, Qingkailing injection, Shenfu injection, Xingnaojing injection, Xueshuantong for injection, Aciclovir sodium, Cefoperazone sodium and Sulbactam sodium (1:1), Ceftriaxone sodium, Cefuroxime sodium, Composite Potassium Hydrogen Phosphate, Ganciclovir sodium, Ilaprazole sodium,

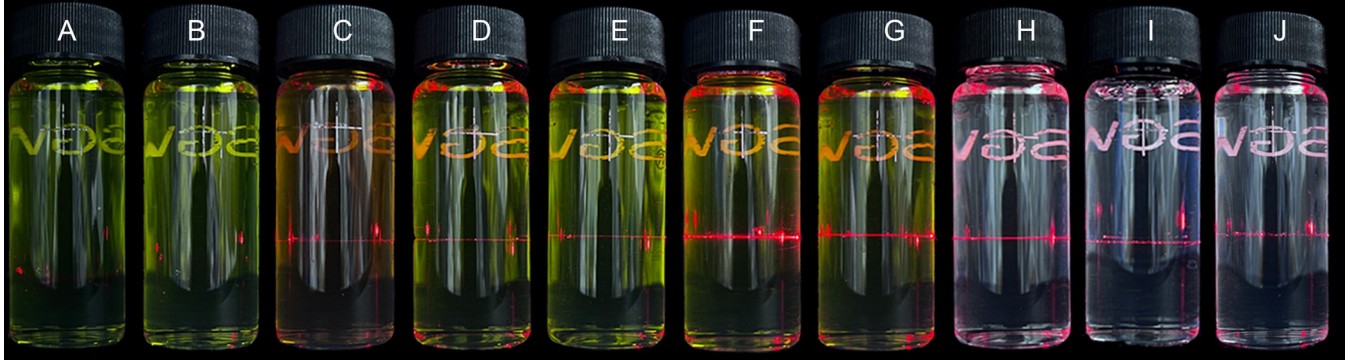

**Fig 2. Tyndall beam of solutions. Legend:** (A) Negative control, XBJ in D5W at 4 h. (B) Negative control, XBJ in NS at 4 h. (C) Combination of XBJ + Acyclovir Sodium in NS at 0 h. (D) Combination of XBJ + Ceftriaxone Sodium in NS at 0 h. (E) Combination of XBJ + Cefuroxime Sodium in NS at 0 h. (F) Combination of XBJ + Iprazole Sodium in NS at 0 h. (G) Combination of XBJ + Rabeprazole Sodium in NS at 0 h. (H) Positive control, Calcium Chloride with Composite Potassium Hydrogen Phosphate in NS at 0 h. (I) Positive control, 10 µm latex particles reference materials. (J) Positive control, 25 µm particle count reference materials.

**Table 3. Results of particle count and pH changes of solutions.**

| Drug | ≥10 μm Particles/ml (SD) | | | | ≥25 μm Particles/ml (SD) | | | | pH and the changes | | | | Compatibility |
|---|---|---|---|---|---|---|---|---|---|---|---|---|---|
| | 0 h | 1 h | 2 h | 4 h | 0 h | 1 h | 2 h | 4 h | 0 h | 1 h | 2 h | 4 h | |
| Aciclovir Sodium | 52.70 (0.68) | 89.01(2.00) | 92.36 (0.89) | 57.44 (4.22) | 2.22 (0.15) | 4.29 (0.25) | 3.29 (0.19) | 3.51 (0.14) | 9.71 | 0.01 | 0.01 | -0.04 | Incomp |
| Ambroxol Hydrochloride | 11.56 (3.74) | 12.22 (10.58) | 11.60 (1.85) | 15.09 (14.11) | 0.29 (0.20) | 0.71 (0.94) | 0.36 (0.10) | 0.36 (0.33) | 5.68 | 0.17 | 0.08 | 0.04 | Comp |
| Calcium Chloride | 6.36 (1.35) | 20.24(1.20) | 10.04 (2.88) | 17.82 (4.17) | 0.29 (0.04) | 1.11 (0.10) | 0.73 (0.18) | 0.49 (0.44) | 5.52 | -0.08 | 0.25 | 0.08 | Comp |
| Calcium Gluconate | 5.33 (5.84) | 6.53(2.17) | 17.49 (8.04) | 5.87 (0.47) | 0.36 (0.08) | 0.11 (0.08) | 1.67 (1.72) | 2.53 (3.01) | 5.22 | 0.04 | 0.02 | 0.05 | Comp |
| Cefazolin Sodium | 10.98 (6.00) | 18.71(3.88) | 18.80 (7.88) | 28.98 (7.47) | 0.31 (0.10) | 0.58 (0.10) | 0.82 (0.48) | 0.80 (0.35) | 5.14 | -0.13 | -0.08 | -0.08 | Incomp |
| Cefoperazone Sodium and Sulbactam Sodium (11) | 53.22 (2.67) | 69.40(2.27) | 43.22 (1.27) | 56.36 (4.35) | 1.02 (0.04) | 2.84 (1.80) | 1.27 (0.12) | 2.33 (0.12) | 4.96 | -0.07 | -0.09 | -0.08 | Incomp |
| Ceftriaxone Sodium | 35.16 (0.60) | 32.07(0.61) | 49.98 (0.50) | 110.93 (1.20) | 0.87 (0.27) | 0.89 (0.17) | 2.04 (0.10) | 3.64 (0.15) | 5.69 | 0.16 | 0.13 | 0.15 | Incomp |
| Cefuroxime Sodium | 32.96 (0.52) | 56.36(0.44) | 114.40 (1.39) | 137.58 (0.43) | 0.60 (0.12) | 1.69 (0.08) | 4.36 (0.34) | 7.53 (0.12) | 5.75 | -0.51 | -0.52 | -0.47 | Incomp |
| Ciwujia injection | 18.38 (0.38) | 20.49(2.87) | 12.98 (2.30) | 14.24 (1.01) | 1.89 (0.10) | 1.02 (0.23) | 0.73 (0.18) | 1.56 (0.08) | 5.16 | -0.02 | 0.00 | 0.03 | Comp |
| Composite Potassium Hydrogen Phosphate | 25.73 (15.00) | 33.04(19.3) | 26.40 (19.54) | 25.29 (6.73) | 0.31 (0.15) | 1.56 (0.14) | 1.62 (2.12) | 0.84 (0.67) | 7.04 | 0.07 | 0.08 | 0.07 | Incomp |
| Danhong injection | 13.56 (2.01) | 11.38(3.33) | 7.64 (3.83) | 19.80 (14.52) | 0.40 (0.13) | 0.87 (0.52) | 0.36 (0.23) | 0.53 (0.35) | 5.77 | -0.11 | 0.00 | -0.05 | Comp |
| Danshen Chuanxiongqin injection | 48.27 (20.09) | 51.98(9.34) | 25.29 (4.39) | 125.53 (27.26) | 1.64 (0.32) | 0.87 (0.31) | 0.51 (0.28) | 5.11 (0.63) | 4.15 | -0.13 | -0.10 | 0.03 | Incomp |
| Dazhu Hongjingtian injection | 3.89 (0.91) | 5.87(0.59) | 21.11 (0.20) | 2.11 (0.23) | 0.91 (0.14) | 0.59 (0.10) | 0.20 (0.14) | 0.23 (0.13) | 5.26 | 0.03 | 0.01 | 0.06 | Comp |
| Dengzhanxixin injection | 40.07 (3.87) | 29.20(1.11) | 37.44 (1.05) | 50.64 (6.58) | 0.56 (0.04) | 0.51 (0.10) | 0.76 (0.04) | 3.73 (0.32) | 5.38 | 0.05 | 0.03 | 0.03 | Incomp |
| Dexamethasone Sodium Phosphate | 11.98 (1.82) | 13.18(6.63) | 7.62 (2.33) | 8.56 (1.65) | 0.33 (0.18) | 0.71 (0.32) | 0.33 (0.12) | 0.20 (0.07) | 6.27 | -0.06 | -0.05 | -0.07 | Comp |
| Doxofylline | 10.80 (6.58) | 13.71(3.93) | 15.02 (5.94) | 9.33 (5.90) | 0.56 (0.44) | 1.18 (0.94) | 0.11 (0.19) | 0.29 (0.14) | 5.92 | 0.23 | -0.09 | -0.12 | Comp |
| Extract of Ginkgo Biloba Leaves | 16.44 (0.17) | 12.78(0.42) | 14.18 (0.37) | 6.02 (1.42) | 1.2 (0.92) | 0.36 (0.14) | 0.64 (0.33) | 0.49 (0.10) | 6.49 | -0.08 | -0.23 | -0.22 | Comp |
| Ganciclovir sodium | 10.58 (0.77) | 67.11(1.44) | 75.22 (1.29) | 35.27 (2.02) | 0.42 (0.23) | 3.56 (0.08) | 3.29 (0.14) | 3.62 (2.21) | 9.31 | 0.01 | 0.05 | -0.33 | Incomp |
| Ginkgo leaf Extract and Dipyridamole | 12.20 (0.00) | 9.02(5.70) | 11.04 (5.96) | 13.22 (3.61) | 0.69 (0.56) | 0.47 (0.64) | 0.80 (0.76) | 0.89 (0.86) | 5.35 | 0.02 | 0.02 | -0.04 | Comp |
| Guanxinning injection | 30.67 (1.67) | 33.28 (10.11) | 44.02 (4.29) | 27.62 (4.53) | 0.80 (0.35) | 0.87 (0.27) | 1.27 (0.47) | 0.98 (0.80) | 5.59 | 0.08 | 0.05 | 0.11 | Incomp |
| Honghua injection | 9.93 (3.82) | 11.31(7.03) | 7.31 (0.04) | 24.04 (5.34) | 0.27 (0.12) | 0.96 (0.60) | 0.58 (0.21) | 1.78 (0.31) | 5.66 | 0.24 | 0.06 | 0.07 | Comp |
| Huangqi injection | 15.36 (3.77) | 10.67(3.47) | 7.49 (2.72) | 15.27 (3.99) | 0.71 (0.39) | 0.51 (0.08) | 0.31 (0.04) | 1.18 (0.20) | 6.71 | -0.18 | -0.18 | -0.26 | Comp |
| HuangqiDuotang for Injection | 19.24 (0.71) | 17.80(0.58) | 11.27 (1.62) | 6.00 (0.69) | 1.67 (0.37) | 0.93 (0.46) | 1.58 (1.00) | 0.22 (0.08) | 5.61 | 0.19 | 0.13 | 0.02 | Comp |
| Hydrocortisone SodiumSuccinate | 16.93 (7.51) | 13.38(8.69) | 7.42 (6.44) | 9.91 (7.19) | 0.69 (0.91) | 0.71 (0.42) | 0.16 (0.21) | 0.67 (0.69) | 6.36 | -0.10 | -0.13 | -0.11 | Comp |
| Ilaprazole Sodium | 56.33 (14.57) | 89.80 (59.03) | 84.76 (12.98) | 37.67 (11.81) | 3.09 (1.89) | 3.36 (3.62) | 3.49 (3.76) | 3.20 (0.80) | 6.37 | -0.04 | -0.11 | -0.09 | Incomp |
| Kang'ai injection | 35.96 (12.29) | 34.36 (19.11) | 29.08 (10.03) | 85.89 (6.93) | 0.89 (0.28) | 0.69 (1.14) | 0.74 (1.00) | 3.67 (1.75) | 6.71 | 0.02 | -0.08 | -0.01 | Incomp |

(*Continued*)

**Table 3.** (Continued)

| Drug | ≥10 μm Particles/ml (SD) | | | | ≥25 μm Particles/ml (SD) | | | | pH and the changes | | | | Compatibility |
|---|---|---|---|---|---|---|---|---|---|---|---|---|---|
| | 0 h | 1 h | 2 h | 4 h | 0 h | 1 h | 2 h | 4 h | 0 h | 1 h | 2 h | 4 h | |
| Kuhuang injection | 66.80 (21.32) | 78.98(0.77) | 43.56 (2.17) | 38.91 (8.61) | 2.20 (0.44) | 0.78 (0.25) | 0.82 (0.19) | 0.62 (0.15) | 5.74 | -0.10 | -0.09 | -0.03 | Incomp |
| Lansoprazole Sodium | 17.00 (0.77) | 13.91(0.28) | 7.44 (0.38) | 9.82 (0.19) | 1.47 (0.13) | 0.80 (0.18) | 0.47 (0.35) | 0.49 (0.04) | 7.69 | 0.05 | 0.02 | -0.09 | Comp |
| Magnesium Isoglycyrrhizinate | 15.60 (7.68) | 10.38(7.18) | 10.51 (5.92) | 13.27 (9.65) | 0.38 (0.23) | 0.40 (0.37) | 0.42 (0.41) | 0.28 (0.09) | 6.21 | 0.05 | -0.07 | 0.10 | Comp |
| Magnesium Sulfate | 12.04 (6.05) | 23.91 (22.46) | 8.33 (4.87) | 8.91 (7.73) | 1.09 (1.35) | 0.31 (0.42) | 0.18 (0.10) | 1.11 (1.75) | 5.62 | 0.01 | -0.21 | -0.23 | Comp |
| Mailuoning injection | 11.98 (1.46) | 17.73 (13.58) | 8.87 (1.92) | 7.42 (0.27) | 0.48 (0.20) | 0.49 (0.34) | 0.31 (0.34) | 0.58 (0.37) | 6.30 | 0.13 | -0.07 | 0.06 | Comp |
| Methylprednisolone Sodium Succibate | 14.87 (6.09) | 23.45 (14.55) | 17.22 (12.65) | 9.02 (6.45) | 0.98 (0.60) | 0.74 (0.63) | 0.98 (0.62) | 0.60 (0.47) | 7.42 | -0.18 | -0.11 | -0.08 | Comp |
| OmeprazoleSodium | 5.07 (1.51) | 7.84(1.81) | 15.11 (3.07) | 10.58 (6.20) | 0.31 (0.14) | 0.38 (0.10) | 0.42 (0.20) | 0.18 (0.17) | 7.94 | 0.05 | 0.08 | 0.02 | Comp |
| Pantoprazole Sodium | 15.69 (9.76) | 8.38(2.04) | 10.27 (4.56) | 7.67 (3.18) | 0.40 (0.29) | 0.18 (0.19) | 0.31 (0.17) | 0.27 (0.29) | 7.33 | 0.13 | 0.18 | 0.05 | Comp |
| Potassium Chloride | 7.44 (1.04) | 4.02(1.65) | 17.40 (3.47) | 14.82 (5.79) | 0.33 (0.18) | 0.18 (0.08) | 1.27 (0.47) | 1.24 (0.93) | 5.35 | 0.36 | 0.22 | 0.49 | Comp |
| Qingkailing injection | 28.58 (1.60) | 29.33(0.48) | 25.78 (3.04) | 30.07 (0.98) | 2.11 (0.23) | 2.00 (0.18) | 1.38 (1.07) | 2.09 (0.04) | 6.24 | 0.08 | 0.18 | -0.01 | Incomp |
| Rabeprazole Sodium | 43.20 (29.84) | 116.22 (110.53) | 93.89 (59.61) | 71.02 (52.03) | 2.67 (1.10) | 5.51 (5.13) | 6.27 (4.91) | 3.04 (2.32) | 6.87 | 0.04 | 0.00 | -0.13 | Incomp |
| Reduced Glutathione | 8.49 (2.33) | 6.67(0.88) | 6.89 (1.83) | 17.71 (7.86) | 0.44 (0.10) | 0.18 (0.17) | 0.31 (0.43) | 0.96 (1.16) | 5.69 | -0.03 | -0.04 | -0.02 | Comp |
| Reduning injection | 9.49 (2.35) | 16.27(1.81) | 10.40 (5.06) | 16.98 (0.99) | 0.69 (0.10) | 0.73 (0.18) | 0.20 (0.07) | 1.07 (0.29) | 4.78 | -0.03 | -0.02 | -0.03 | Comp |
| Shenfu injection | 33.31 (2.49) | 30.53(1.83) | 88.6 (6.4) | 87.47 (4.67) | 1.62 (0.19) | 0.44 (0.10) | 1.53 (0.13) | 2.82 (0.15) | 5.79 | 0.08 | 0.08 | 0.19 | Incomp |
| Shenmai injection | 8.76 (0.34) | 8.98(4.52) | 4.07 (1.77) | 18.96 (3.74) | 0.27 (0.07) | 0.56 (0.14) | 0.36 (0.15) | 0.53 (0.29) | 5.25 | 0.02 | 0.06 | 0.10 | Comp |
| Shenqi Fuzheng injection | 6.62 (4.31) | 11.16(1.46) | 16.84 (0.62) | 6.84 (1.08) | 0.11 (0.19) | 0.53 (0.12) | 0.93 (0.58) | 0.53 (0.46) | 5.58 | 0.23 | 0.00 | 0.03 | Comp |
| Shuanghuanglian injection | 17.07 (0.47) | 19.53(0.57) | 11.11 (1.44) | 15.47 (0.84) | 0.89 (0.17) | 1.24 (0.08) | 0.42 (0.14) | 0.76 (0.17) | 5.21 | 0.10 | 0.10 | -0.01 | Comp |
| Shuxuetong injection | 6.91 (2.57) | 9.62(0.47) | 6.76 (2.08) | 7.60 (2.48) | 0.58 (0.10) | 0.56 (0.20) | 0.36 (0.10) | 0.33 (0.13) | 5.14 | -0.01 | 0.00 | 0.03 | Comp |
| Sulfotanshinone sodium | 8.80 (2.27) | 4.73(2.50) | 9.67 (2.66) | 11.73 (4.97) | 0.44 (0.21) | 0.18 (0.14) | 0.27 (0.18) | 0.49 (0.44) | 5.50 | 0.03 | 0.01 | -0.01 | Comp |
| Tanreqing injection | 14.87 (4.47) | 12.09(2.14) | 12.8 (1.44) | 41.16 (7.02) | 0.38 (0.20) | 0.42 (0.17) | 0.49 (0.17) | 1.84 (0.57) | 6.40 | 0.20 | 0.02 | 0.02 | Incomp |
| Xiangdan injection | 17.76 (6.80) | 14.38(3.22) | 10.69 (0.74) | 14.44 (5.35) | 1.27 (0.72) | 0.67 (0.57) | 0.29 (0.04) | 0.69 (0.27) | 5.57 | 0.14 | 0.14 | 0.09 | Comp |
| Xingnaojing injection | 9.87 (3.03) | 6.27(3.23) | 51.73 (23.98) | 39.33 (4.24) | 0.42 (0.14) | 0.87 (0.72) | 2.60 (0.75) | 2.96 (0.21) | 5.73 | -0.03 | -0.23 | -0.20 | Incomp |
| Xinmailong injection | 13.31 (1.85) | 6.47(4.72) | 7.60 (3.07) | 8.18 (1.10) | 0.73 (0.12) | 0.40 (0.20) | 0.20 (0.07) | 0.29 (0.17) | 5.64 | -0.08 | -0.10 | 0.06 | Comp |
| Yinxingneizhi injection | 19.24 (0.92) | 16.16(0.76) | 20.07 (0.29) | 13.04 (1.77) | 1.24 (0.04) | 1.11 (0.10) | 0.93 (0.00) | 1.31 (0.08) | 5.51 | -0.05 | 0.01 | 0.10 | Comp |
| Xiyanping injection | 7.56 (2.27) | 11.69(4.85) | 12.36 (11.71) | 7.60 (2.91) | 0.38 (0.25) | 0.31 (0.37) | 0.18 (0.25) | 0.13 (0.00) | 5.73 | -0.04 | -0.05 | -0.19 | Comp |
| Xuesaitong injection | 8.36 (0.91) | 18.58(7.71) | 14.73 (0.66) | 14.96 (5.63) | 0.38 (0.17) | 1.62 (0.25) | 1.11 (0.10) | 0.84 (0.44) | 5.41 | 0.17 | 0.26 | 0.12 | Comp |

*(Continued)*

**Table 3.** (Continued)

| Drug | ≥10 μm Particles/ml (SD) | | | | ≥25 μm Particles/ml (SD) | | | | pH and the changes | | | | Compatibility |
|---|---|---|---|---|---|---|---|---|---|---|---|---|---|
| | 0 h | 1 h | 2 h | 4 h | 0 h | 1 h | 2 h | 4 h | 0 h | 1 h | 2 h | 4 h | |
| Xueshuantong for injection | 53.13 (6.13) | 27.87(2.08) | 27.02 (0.28) | 46.73 (4.20) | 1.87 (0.07) | 0.71 (0.10) | 1.42 (0.08) | 2.40 (0.20) | 5.06 | 0.03 | -0.14 | 0.01 | Incomp |
| XBJ[A] | 20.11 (2.93) | 10.87(0.31) | 15.51 (0.30) | 8.60 (0.24) | 0.64 (0.37) | 0.93 (0.35) | 1.51 (0.51) | 0.73 (0.13) | 5.53 | 0.02 | 0.01 | 0.03 | Comp |
| XBJ[B] | 13.98 (2.54) | 18.67(1.65) | 14.29 (0.57) | 20.33 (0.42) | 0.44 (0.20) | 1.56 (0.34) | 1.53 (0.37) | 0.96 (0.25) | 6.26 | 0.00 | 0.00 | 0.00 | Comp |
| Calcium Chloride with Composite Potassium Hydrogen Phosphate[C] | 192.07 (56.11) | 41257.73 (499.97) | - | - | 17.47 (14.40) | 146.60 (12.69) | - | - | - | | | | Incomp |
| 10 μm latex particles reference material[D] | 1236(4.20) | | | | - | | | | - | | | | Incomp |
| 25 μm particle count reference material[E] | - | | | | 1973(3.87) | | | | - | | | | Incomp |

**Note:** (A) Negative control, XBJ (Xuebijing injection) in D5W (dextrose 5% water). (B) Negative control, XBJ in NS (Normal saline). (C) Positive control, Calcium Chloride with Composite Potassium in NS. (D) Positive control, 10 μm latex particles reference material. (E) Positive control, 25 μm particle count reference material, Comp; compatible, Incomp; incompatible

Rabeprazole sodium, respectively. Then showed a increase within the 0 to 4 h in our study. The particles in above solutions exceed the recommended the limitation of the Ch.P specification. At the time of 4 h particles in group of XBJ + Tanreqing injection and XBJ + Cefazolin sodium also exceed the limitation of the Ch.P, indicating that XBJ was incompatible with above 19 drugs.

### 3.4 pH changes findings

Table 3 exhibited pH range of combinations of XBJ with other intravenous drugs (4.0 to 10.0). pH changes that did not vary >10% from baseline (immediately after mixing) in any admixture.

### 3.5 Chromacity value changes findings

Table 4 exhibited XBJ had no change in color visually and chromacity value ≥700 within 4 h with the selected 53 intravenous drugs mixing solutions. It inferred that the chromacity value change of 1, 2, 4 h were <200 compared to 0 h, two drugs were physically compatible.

### 3.6 Photometrical changes findings

Absorption at 550 nm were detected. Binary combinations of XBJ with Danshen Chuanxiong-qin injection, Guanxinning injection, Kang'ai injection, Kuhuang injection, Tanreqing injection, Xiangdan injection, Xueshuantong for injection, Ceftriaxone sodium, Cefuroxime sodium, Ganciclovir sodium, Ilaprazole sodium, Lansoprazole sodium, Omeprazole sodium, Pantoprazole sodium, Rabeprazole sodium solutions immediately displayed $A_{550 nm}$ changes >0.0100 compared to 0 h and combinations of XBJ with Extract of Ginkgo Biloba Leaves, Qingkailing injection, Yinxingneizhi injection, Aciclovir sodium, Doxofylline, Magnesium Sulfate solutions displayed $A_{550 nm}$ changes >0.0100 at 4 h compared to 0 h (Table 4), the results showed that XBJ was incompatible with above 21 drugs.

Binary combinations of XBJ with 53 intravenous drug solutions were primarily yellow or brown-yellow in appearance. The range of $A_{420 nm}$ varied from 2.8000 to 4.1000, resulting in a significant error. The change in $A_{420 nm}$ exceeded 0.0400 (Table 4).

**Table 4. Results of chromacity value, photometrical changes of solutions.**

| Drug | Chromacity value | | | | A$_{550\,nm}$ | | | | A$_{420\,nm}$ | | | | Compatibility |
|---|---|---|---|---|---|---|---|---|---|---|---|---|---|
| ' | 0 h | 1 h | 2 h | 4 h | 0 h | 1 h | 2 h | 4 h | 0 h | 1 h | 2 h | 4 h | |
| Aciclovir Sodium | ≥700 | ≥700 | ≥700 | ≥700 | 0.1613 | -0.0030 | -0.0034 | -0.0109 | 3.4149 | 0.3867 | -0.1297 | -0.2893 | Incomp |
| Ambroxol Hydrochloride | ≥700 | ≥700 | ≥700 | ≥700 | 0.0080 | 0.0023 | 0.0025 | 0.0058 | 3.4153 | 0.0188 | -0.0082 | -0.2460 | Comp |
| Calcium Chloride | ≥700 | ≥700 | ≥700 | ≥700 | 0.0270 | -0.0016 | 0.0037 | -0.0012 | 3.0196 | 0.1606 | 0.2599 | 0.0280 | Comp |
| Calcium Gluconate | ≥700 | ≥700 | ≥700 | ≥700 | 0.0377 | 0.0030 | -0.0026 | 0.0013 | 3.1794 | 0.1750 | -0.0898 | -0.0565 | Comp |
| Cefazolin Sodium | ≥700 | ≥700 | ≥700 | ≥700 | 0.0386 | -0.0078 | -0.0084 | -0.0016 | 3.0707 | 0.4705 | 0.3936 | -0.1086 | Comp |
| Cefoperazone Sodium and Sulbactam Sodium (11) | ≥700 | ≥700 | ≥700 | ≥700 | 0.0300 | 0.0032 | -0.0015 | 0.0040 | 3.0079 | 0.2756 | 0.4249 | 0.9797 | Comp |
| Ceftriaxone Sodium | ≥700 | ≥700 | ≥700 | ≥700 | 0.0518 | -0.0144 | -0.0139 | -0.0391 | 3.2581 | -0.0954 | -0.3711 | -0.1984 | Incomp |
| Cefuroxime Sodium | ≥700 | ≥700 | ≥700 | ≥700 | 0.0366 | 0.0117 | -0.0132 | -0.0353 | 3.1501 | 0.5438 | 0.0161 | -0.1463 | Incomp |
| Ciwujia injection | ≥700 | ≥700 | ≥700 | ≥700 | 0.1388 | -0.0067 | -0.0004 | 0.0026 | 4.0643 | -0.6629 | -0.8177 | -0.9129 | Comp |
| Composite Potassium Hydrogen Phosphate | ≥700 | ≥700 | ≥700 | ≥700 | 0.0404 | 0.0034 | -0.0005 | 0.0089 | 3.2262 | -0.0030 | 0.2350 | 0.0297 | Comp |
| Danhong injection | ≥700 | ≥700 | ≥700 | ≥700 | 0.0771 | -0.0010 | -0.0087 | -0.0092 | 3.3121 | 0.1410 | 0.2379 | -0.1409 | Comp |
| Danshen Chuanxiongqin injection | ≥700 | ≥700 | ≥700 | ≥700 | 0.0171 | 0.0040 | 0.0222 | -0.0148 | 3.1010 | 0.0152 | 0.1192 | 0.2142 | Incomp |
| Dazhu Hongjingtian injection | ≥700 | ≥700 | ≥700 | ≥700 | 0.0295 | 0.0033 | 0.0015 | 0.0024 | 3.3764 | -0.1408 | -0.2946 | -0.2731 | Comp |
| Dengzhanxixin injection | ≥700 | ≥700 | ≥700 | ≥700 | 0.0556 | 0.0005 | -0.0003 | -0.0036 | 3.2579 | -0.1641 | -0.2153 | -0.0647 | Comp |
| Dexamethasone Sodium Phosphate | ≥700 | ≥700 | ≥700 | ≥700 | 0.0209 | -0.0007 | 0.0002 | -0.0015 | 3.0749 | 0.0419 | 0.2617 | 0.1608 | Comp |
| Doxofylline | ≥700 | ≥700 | ≥700 | ≥700 | 0.0113 | 0.0008 | 0.0063 | 0.0116 | 3.1805 | 0.2562 | 0.1806 | -0.1239 | Incomp |
| Extract of Ginkgo Biloba Leaves | ≥700 | ≥700 | ≥700 | ≥700 | 0.0468 | 0.0039 | -0.0090 | -0.0132 | 3.1753 | 0.6972 | 0.6402 | 0.2115 | Incomp |
| Ganciclovir sodium | ≥700 | ≥700 | ≥700 | ≥700 | 0.1296 | 0.0137 | 0.0148 | 0.0198 | 3.6969 | -0.4958 | -0.8039 | -0.9113 | Incomp |
| Ginkgo leaf Extract and Dipyridamole | ≥700 | ≥700 | ≥700 | ≥700 | 0.0292 | -0.0090 | -0.0016 | 0.0012 | 3.4612 | -0.1975 | -0.2302 | -0.2526 | Comp |
| Guanxinning injection | ≥700 | ≥700 | ≥700 | ≥700 | 0.0531 | 0.0115 | 0.0200 | 0.0276 | 3.4168 | -0.1032 | -0.0110 | -0.1160 | Incomp |
| Honghua injection | ≥700 | ≥700 | ≥700 | ≥700 | 0.0668 | -0.0093 | -0.0007 | 0.0029 | 3.5088 | -0.0843 | -0.2431 | -0.2447 | Comp |
| Huangqi injection | ≥700 | ≥700 | ≥700 | ≥700 | 0.0564 | 0.0062 | -0.0098 | 0.0013 | 3.4831 | -0.3366 | -0.2205 | -0.3444 | Comp |
| Huangqi Duotang for Injection | ≥700 | ≥700 | ≥700 | ≥700 | 0.0265 | 0.0099 | -0.0040 | -0.0006 | 3.1719 | 0.1836 | 0.0303 | 0.1164 | Comp |
| Hydrocortisone Sodium Succinate | ≥700 | ≥700 | ≥700 | ≥700 | 0.0323 | 0.0004 | 0.0036 | 0.0065 | 3.2429 | 0.2774 | 0.2582 | 0.1278 | Comp |
| Ilaprazole Sodium | ≥700 | ≥700 | ≥700 | ≥700 | 0.0514 | 0.0245 | 0.0327 | 0.0407 | 3.1529 | 0.2611 | 0.6093 | 0.0532 | Incomp |
| Kang'ai injection | ≥700 | ≥700 | ≥700 | ≥700 | 0.0189 | 0.0129 | 0.0202 | 0.0125 | 3.2428 | -0.1354 | -0.1632 | -0.1913 | Incomp |
| Kuhuang injection | ≥700 | ≥700 | ≥700 | ≥700 | 0.0517 | 0.0250 | 0.0244 | 0.0285 | 3.1223 | -0.0929 | -0.0834 | -0.0620 | Incomp |
| Lansoprazole | ≥700 | ≥700 | ≥700 | ≥700 | 0.0361 | 0.0125 | 0.0229 | 0.0177 | 3.6358 | -0.1696 | -0.0175 | 0.0663 | Incomp |
| Magnesium Isoglycyrrhizinate | ≥700 | ≥700 | ≥700 | ≥700 | 0.0184 | -0.0022 | -0.0042 | -0.0023 | 3.3400 | 0.0524 | 0.0403 | -0.0604 | Comp |
| Magnesium Sulfate | ≥700 | ≥700 | ≥700 | ≥700 | 0.0308 | 0.0030 | 0.0082 | 0.0106 | 3.1031 | 0.1857 | 0.2559 | 0.0463 | Incomp |
| Mailuoning injection | ≥700 | ≥700 | ≥700 | ≥700 | 0.0505 | 0.0040 | 0.0018 | -0.0052 | 3.4737 | 0.3838 | 0.3483 | 0.0360 | Comp |
| Methylprednisolone Sodium Succibate | ≥700 | ≥700 | ≥700 | ≥700 | 0.0455 | -0.0002 | 0.0025 | 0.0042 | 3.4922 | 0.1789 | -0.0243 | -0.0214 | Comp |
| Omeprazole Sodium | ≥700 | ≥700 | ≥700 | ≥700 | 0.0573 | -0.0228 | -0.0169 | 0.0130 | 3.5021 | -0.2590 | -0.3366 | -0.4169 | Incomp |
| Pantoprazole Sodium | ≥700 | ≥700 | ≥700 | ≥700 | 0.0517 | -0.0227 | -0.0260 | -0.0115 | 3.4664 | -0.0328 | -0.2284 | -0.3333 | Incomp |
| Potassium Chloride | ≥700 | ≥700 | ≥700 | ≥700 | 0.0156 | 0.0028 | -0.0016 | -0.0008 | 2.9599 | 0.1368 | 0.1969 | 0.0247 | Comp |
| Qingkailing injection | ≥700 | ≥700 | ≥700 | ≥700 | 0.0486 | 0.0024 | 0.0002 | 0.0115 | 3.3089 | -0.0117 | -0.0236 | -0.1111 | Incomp |
| Rabeprazole Sodium | ≥700 | ≥700 | ≥700 | ≥700 | 1.1166 | 0.4382 | 1.0674 | 1.8095 | 3.1098 | 0.2287 | 0.2870 | 0.4238 | Incomp |
| Reduced Glutathione | ≥700 | ≥700 | ≥700 | ≥700 | 0.0299 | 0.0031 | -0.0068 | -0.0042 | 3.4335 | 0.0052 | 0.0303 | -0.1220 | Comp |
| Reduning injection | ≥700 | ≥700 | ≥700 | ≥700 | 0.0596 | 0.0047 | 0.0057 | 0.0075 | 3.3104 | 0.5311 | 0.3123 | 0.1935 | Comp |
| Shenfu injection | ≥700 | ≥700 | ≥700 | ≥700 | 0.0407 | -0.0089 | -0.0058 | -0.0009 | 3.2368 | 0.0442 | 0.0310 | -0.0339 | Comp |
| Shenmai injection | ≥700 | ≥700 | ≥700 | ≥700 | 0.0446 | 0.0018 | 0.0016 | -0.0014 | 3.2116 | 0.3510 | 0.5023 | -0.0759 | Comp |
| Shenqi Fuzheng injection | ≥700 | ≥700 | ≥700 | ≥700 | 0.0312 | -0.0025 | -0.0046 | -0.0002 | 3.2320 | 0.7015 | 0.3855 | 0.1261 | Comp |
| Shuanghuanglian injection | ≥700 | ≥700 | ≥700 | ≥700 | 0.0834 | 0.0022 | -0.0072 | 0.0024 | 3.6586 | -0.0024 | 0.2875 | -0.0681 | Comp |
| Shuxuetong injection | ≥700 | ≥700 | ≥700 | ≥700 | 0.0475 | -0.0047 | -0.0032 | -0.0037 | 3.2062 | 0.5964 | 0.2515 | 0.0008 | Comp |
| Sulfotanshinone sodium | ≥700 | ≥700 | ≥700 | ≥700 | 0.2261 | 0.0069 | 0.0043 | 0.0032 | 3.8027 | -0.4262 | -0.4684 | -0.6765 | Comp |

*(Continued)*

**Table 4.** (Continued)

| Drug | Chromacity value | | | | A$_{550\ nm}$ | | | | A$_{420\ nm}$ | | | | Compatibility |
|---|---|---|---|---|---|---|---|---|---|---|---|---|---|
| ' | 0 h | 1 h | 2 h | 4 h | 0 h | 1 h | 2 h | 4 h | 0 h | 1 h | 2 h | 4 h | |
| Tanreqing injection | ≥700 | ≥700 | ≥700 | ≥700 | 0.0767 | 0.0139 | -0.0308 | 0.0119 | 3.3278 | 0.2529 | 0.1107 | -0.0549 | Incomp |
| Xiangdan injection | ≥700 | ≥700 | ≥700 | ≥700 | 0.0537 | -0.0055 | -0.0219 | -0.0184 | 3.2809 | -0.0320 | -0.0320 | -0.0521 | Incomp |
| Xingnaojing injection | ≥700 | ≥700 | ≥700 | ≥700 | 0.0264 | 0.0076 | 0.0078 | 0.0049 | 3.2257 | 0.3063 | 0.2352 | 0.2463 | Comp |
| Xinmailong injection | ≥700 | ≥700 | ≥700 | ≥700 | 0.0312 | -0.0092 | -0.0033 | 0.0005 | 3.2165 | 0.2327 | 0.3796 | -0.0130 | Comp |
| Yinxingneizhi injection | ≥700 | ≥700 | ≥700 | ≥700 | 0.0291 | 0.0039 | 0.0082 | 0.0136 | 3.1696 | -0.0444 | -0.0700 | -0.0423 | Incomp |
| Xiyanping injection | ≥700 | ≥700 | ≥700 | ≥700 | 0.0258 | 0.0017 | 0.0056 | 0.0009 | 3.2403 | 0.6899 | 0.4666 | 0.0525 | Comp |
| Xuesaitong injection | ≥700 | ≥700 | ≥700 | ≥700 | 0.0323 | 0.0038 | 0.0066 | 0.0028 | 3.2665 | 0.2646 | 0.0040 | -0.0851 | Comp |
| Xueshuantong for injection | ≥700 | ≥700 | ≥700 | ≥700 | 0.0080 | 0.0414 | 0.0512 | 0.0525 | 2.8011 | 0.0578 | 0.0425 | 0.2291 | Incomp |
| XBJ[A] | ≥700 | ≥700 | ≥700 | ≥700 | 0.0875 | -0.0058 | -0.0097 | -0.0087 | 3.3454 | 0.3226 | 0.2888 | 0.3435 | Comp |
| XBJ[B] | ≥700 | ≥700 | ≥700 | ≥700 | 0.0581 | -0.0023 | -0.0043 | 0.0032 | 3.5740 | -0.0799 | -0.2953 | -0.3025 | Comp |

**Note:** (A) Negative control, XBJ (Xuebijing injection) in D5W (dextrose 5% water). (B) Negative control, XBJ in NS (Normal saline), Comp; compatible, Incomp; incompatible

## 4. Discussion

XBJ infusion was usually administered clinically within 4 hours. In this study, the physical compatibility for XBJ in binary combinations with selected 53 intravenous drugs were evaluated. Table 2 showed that no visual changes (color changes, gas evolution, haze, or visible particulate formation) were detected over the 4 hours period for the solutions. However, the combination of XBJ +Acyclovir solutions, XBJ + Ceftriaxone sodium, XBJ + Cefuroxime sodium, XBJ + Iprazole sodium and XBJ + Rabeprazole sodium solutions in NS produced Tyndall Beam respectively (Fig 2), indicating that XBJ was incompatible with above 5 drugs. Since the discovery of the Tyndall effect, it has been mainly used to assist the observation of the effect of colloidal solutions in experiments. As a control index of solution properties for intravenous drugs, Tyndall effect can directly reflect the solution quality and physical compatibility of intravenous drugs without damaging the package of drug solutions [11, 12, 17]. In our study, three inspections of color change, clarification and Tyndall effect were used to evaluate the properties of solutions, and it was compared with 10 μm latex particles material, 25 μm particle count reference material and mixed solution of Calcium Chloride with Composite Potassium, which could produce Tyndall effect. It was proved that Tyndall effect could be used as an inspection method of solution properties and physical compatibility of Chinese medicine injection. Three inspections can objectively reflect the quality of the solution of Chinese medicine injection. However, our studies found that oily Chinese medicine fat milk injections such as Brucea Javanica oil emulsion injection, Elemene injectable emulsion and Zedoray Turmeric oil injection can produce Tyndall beam, which is not suitable for this inspection method. The inspection of these kind of intravenous drug solutions need further study in our research.

The turbidity of the solutions can be measured by a turbidity meter. Different size and qualitative of particle matter in solutions can scatter the incident light. The turbidity of the solutions can be checked by measuring the intensity of the transmitted or scattered light. The transmitted-scattered light comparative measurement model is used for the turbidity determination of low and medium turbidity colorless solutions (turbidity below 100 NTU). The colored substance may reduce the turbidity of the solution, but yellow has the least effect on the results [18]. The diluted solution of XBJ was a yellow solution, 31 kinds of traditional Chinese medicine injection were mainly yellow clarified liquid, 22 kinds of intravenous drug solutions were colorless or almost colorless clarified liquid. The combination solutions of XBJ with other

53 drugs solutions were mainly yellow or browish-yellow clarified liquid, which was also suitable for turbidity determination. The turbidity change of the solution can avoid the possible influence of color on turbidity, and physical incompatibility was defined as $\geq$0.5 NTU change in turbidity compared to 0 h [13, 14]. From above tests we found that XBJ was incompatible with Iprrazole sodium and Rabeprazole sodium.

Part 4 of the Ch.P recommends that injectable solutions be analyzed using a light obscuration particle count test [10]. For this study, XBJ with Danshen Chuanxiongqin injection, Dengzhanxixin injection, Guanxinning injection, Kang'ai injection, Kuhuang injection, Qingkailing injection, Shenfu injection, Xingnaojing injection, Xueshuantong for injection, Aciclovir sodium, Cefoperazone sodium and Sulbactam sodium (1:1), Ceftriaxone sodium, Cefuroxime sodium, Composite Potassium Hydrogen Phosphate, Ganciclovir sodium, Ilaprazole sodium, Rabeprazole sodium solutions result in particles exceed the limitation of the Ch.P immediately after mixing and also after 4 hours. The particles in XBJ + Tanreqing injection and XBJ + Cefazolin sodium solutions exceed the limitation of the Ch.P at the time of 4 h, indicating that XBJ was incompatible with above 19 drugs. Interestingly, Tyndall Beam presented in the combination of XBJ + Acyclovir sodium, XBJ + Ceftriaxone sodium, XBJ + Cefuroxime sodium, XBJ + Iprazole sodium and XBJ + Rabeprazole sodium groups, indicated that exceed the limitation of particles might produce Tyndall Beam. During the test, XBJ and select drug were slowly diluted in NS or D5W respectively, mixed slowly along the bottle wall to avoid air bubbles affecting the count of particles. If bubbles are formed, solutions were recommended to stand briefly or subjected to ultrasonic defoaming for 1 min. It was found that XBJ diluted with NS or D5W with concentration $\geq$0.5 ml/ml might result in exceeding the limitation of particles and produce Tyndall Beam. Thus, it was recommended to use XBJ concentration $\leq$0.33 ml/ml clinically. Our pre-experiment found that particles counts were not suitable for the quality control of oily Chinese medicine injectable emulsion such as Brucea Javanica oil emulsion injection, Elemene injectable emulsion and Zedoray Turmeric oil injection, which were suitable for "Determination of particle size and particle size distribution method". The further studies will conduct in next stage.

Studies reported that pH changes of intravenous drug solution within the range of 0.2 to 1.0, the solution was stable and compatible [19–22]. Dotson et al. defined physical incompatibility as a pH value of no more than 10% change from baseline (0 hour) after 48 hours [15]. For our pH measurements (Table 4), all solutions were within the range of 4.0 to 10.0. Compared with 0h, if the pH value of the solution changes by 10%, the pH result will change within 0.4~1.0 accordingly. We intentionally to select the stricter pH changes defined as <10% to confer compatibility in our study, and all solutions had pH change <10% over the time of test. Compare to visual measurement, using a colorimeter to determine the chromacity value of the solutions have advantages in accurately and quantitatively. However, turbid solutions, viscosity solutions or fluorescent solutions will affect the transmission of the inspection light, they are not suitable for chromacity value measurement. Binary combinations of XBJ with selected 53 intravenous drug solutions are clear, non-viscous, non-fluorescent liquid, so it is suitable for chromacity value determination. From results in Table 4, chromacity value of all solutions were $\geq$700 and had no change in color visually with 4 h. We inferred that the chromacity value changes are less than 200, two drugs were considered physically compatible. Based on previously published literatures, colorless, almost colorless or lightly colored drug solutions were considered compatible if $A_{420\ nm}$ <0.0400 [16, 19]. Generally, the absorbance of test solution falls between 0.3 and 0.7, indicating a small margin of measurement error. Binary combinations of XBJ with 53 intravenous drug solutions were primarily yellow or brown-yellow in appearance. The range of $A_{420\ nm}$ varied from 2.8000 to 4.1000, resulting in a significant error. The change in $A_{420\ nm}$ exceeded 0.0400 (Table 4). The results indicated that criterion of $A_{420}$

$_{nm}$ was not suitable for physically compatibility of our study. The light transmittance of water for injection at $A_{550\ nm}$ is 100% and the absorbance is 0. Studies reported that drug solutions were considered compatible if $A_{550\ nm} < 0.0100$ [16, 19]. 21 drugs were incompatible with XBJ during the test.

Due to the lack knowledge on drug incompatibility issues and ways on how to avoid them, incompatibility is often under-recognised by health care practitioners. Incorrect processing with incompatibility events will bring consequences for workload, infection and cost [23]. How to handle y-site the situation need more studies to explore and verify. However, when co-administration is inevitable, flushing or filter is needed [24]. Chinese medicine injections primarily consist of water extracts from complex Chinese herbs. During storage and application, the insoluble particles can increase, which may pose potential risks. To ensure the safety of these injections, it is advisable to use a disposable precision filter infusion set with a 5.0 μm diameter before intravenous infusion. Additionally, proper storage conditions should be maintained to prevent contamination by microorganisms, ensuring the physical, chemical, and pharmacodynamic stability of the drugs. However, chemical and pharmacodynamic measurements are time-consuming and limited by instruments and experimental conditions, which do not meet the requirements for obtaining rapid results. Therefore, the evaluation of Chinese medicine injections should primarily focus on physical compatibility criteria, including visual inspection, Tyndall beam, turbidity, particles, pH, chromacity, and absorption at A550 nm. Nevertheless, considering chemical compatibility can provide further insights into the quality of the solutions.

## 5. Conclusions

The study clearly illustrated the physical compatibility of combinations of XBJ with selected 53 intravenous drugs. Of the 53 tested drugs, our findings demonstrated that XBJ was physically incompatible with 27 intravenous drugs, including 13 Chinese medicine injections: Danshen Chuanxiongqin injection, Dengzhanxixin injection, Extract of Ginkgo Biloba Leaves, Guanxinning injection, Kang'ai injection, Kuhuang injection, Qingkailing injection, Shenfu injection, Tanreqing injection, Xiangdan injection, Xingnaojing injection, Yinxingneizhi injection, Xueshuantong for injection,14 chemical drugs: Aciclovir sodium, Cefazolin sodium, Cefuroxime sodium, Ceftriaxone sodium, Cefoperazone sodium and Sulbactam sodium (1:1), Composite Potassium Hydrogen Phosphate injection, Doxofylline, Ganciclovir sodium, Ilaprazole sodium, Lansoprazole sodium, Magnesium Sulfate, Omeprazole sodium, Pantoprazole sodium and Rabeprazole sodium (Table 5). A total of 26 drugs were compatible with XBJ. XBJ should

**Table 5. Description of XBJ physical incompatibilities with 27 drugs.**

| Drug | Time after mixing with XBJ | | | |
|---|---|---|---|---|
| | Immediately | 1 h | 2 h | 4 h |
| Aciclovir Sodium | a, b | a, b, c | a, b, c | a, b, c, d |
| Cefazolin Sodium | - | - | - | b |
| Cefoperazone Sodium and Sulbactam Sodium (11) | b | b | b | b |
| Ceftriaxone Sodium | a, b | a, b, d | a, b, d | a, b, c, d |
| Cefuroxime Sodium | b | b, d | a, b, c, d | a, b, c, d |
| Composite Potassium Hydrogen Phosphate | b | b | b | b |
| Danshen Chuanxiongqin injection | b | b | b, d | b, c, d |
| Dengzhanxixin injection | b | b | b | b, c |
| Doxofylline | - | - | - | d |

(*Continued*)

**Table 5.** (Continued)

| Drug | Time after mixing with XBJ | | | |
|---|---|---|---|---|
| | Immediately | 1 h | 2 h | 4 h |
| Extract of Ginkgo Biloba Leaves | - | - | - | d |
| Ganciclovir sodium | - | b, c, d | b, c, d | b, c, d |
| Guanxinning injection | b | b, d | b, d | b, d |
| Ilaprazole Sodium | a, b, c | a, b, c, d, e | a, b, c, d, e | a, b, c, d, e |
| Kang'ai injection | b | b, d | b, d | b, c, d |
| Kuhuang injection | b | b, d | b, d | b, d |
| Lansoprazole | - | d | d | d |
| Magnesium Sulfate | - | - | - | d |
| Omeprazole Sodium | - | d | d | d |
| Pantoprazole Sodium | - | d | d | d |
| Qingkailing injection | b | b | b | b, d |
| Rabeprazole Sodium | a, b, c | a, b, c, d, e | a, b, c, d, e | a, b, c, d, e |
| Shenfu injection | b | b | b | b |
| Tanreqing injection | - | d | d | b, d |
| Xiangdan injection | - | - | d | d |
| Xingnaojing injection | - | - | b | b |
| Yinxingneizhi injection | - | - | - | d |
| Xueshuantong for injection | b | b, d | b, d | b, d |

**Note:** Tyndall positive = a, $\geq$10 μm particles exceed 25 particles/ml = b, $\geq$25 μm particles exceed 3 particles/ml = c, change of $A_{550\ nm}$ $\geq$0.0100 = d, turbidity increased by $\geq$0.5NTU = e

not be simultaneously co-administered with above 27 drugs through the Y tube. If coadministration is necessary, it is recommended to flush the infusion tube with an appropriate amount of NS or D5W before and after infusion of XBJ. The novel findings on the physical compatibility of XBJ broadens current knowledge of the safe coadministration of intravenous drugs clinically. It provides a safer infusion in ward. Additionally, we provide scientific and feasible methods for compatibility testing of Chinese herbal injection with other intravenous drugs for the quality control of infusion. These findings have significant implications for clinical practice.

## Supporting information

**S1 Raw data.**
(XLSX)

## Acknowledgments

We thank all participants and their families for their support and assistance. We also wish to thank the following people for their invaluable help in this study: Ning He, Jianglei Xiong, Dongtao Lu, Xuan Nie, Shuo Zhou, Yunlong Zhao and Zhen Zhou.

## Author Contributions

**Conceptualization:** Tong Tong, Peifang Li, Sheng Liu.

**Data curation:** Peifang Li, Haiwen Ding.

**Methodology:** Peifang Li.

**Supervision:** Ying Huang, Sheng Liu.

**Visualization:** Peifang Li.

**Writing – original draft:** Tong Tong.

**Writing – review & editing:** Sheng Liu.

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
