## [Decision Letter · Decision Letter 0]

19 Dec 2023

PONE-D-23-34697Physical compatibility of Xuebijing injection with 53 intravenous drugs during simulated Y-site administrationPLOS ONE

Dear Dr. Liu,

Thank you for submitting your manuscript to PLOS ONE. After careful consideration, we feel that it has merit but does not fully meet PLOS ONE’s publication criteria as it currently stands. Therefore, we invite you to submit a revised version of the manuscript that addresses the points raised during the review process.

We look forward to receiving your revised manuscript.

Kind regards,

Adeel Sattar, Ph.D

Academic Editor

PLOS ONE

Journal Requirements:

2. In the online submission form, you indicated that "The datasets generated or analysed during this study are available from the corresponding author on reasonable request." 

3. Please include your tables as part of your main manuscript and remove the individual files. Please note that supplementary tables (should remain/ be uploaded) as separate ""supporting information"" files.

Additional Editor Comments:

According to reviewers comments and my personal view, authors need to present their data in a more attractive way for readers. Please consider reviewers suggestions and interpret your data in proper way.

Reviewers' comments:

Reviewer's Responses to Questions

**Comments to the Author**

1. Is the manuscript technically sound, and do the data support the conclusions?

Reviewer #1: Yes

Reviewer #2: Partly

2. Has the statistical analysis been performed appropriately and rigorously? 

Reviewer #1: Yes

Reviewer #2: Yes

3. Have the authors made all data underlying the findings in their manuscript fully available?

Reviewer #1: Yes

Reviewer #2: No

4. Is the manuscript presented in an intelligible fashion and written in standard English?

Reviewer #1: Yes

Reviewer #2: No

5. Review Comments to the Author

Reviewer #1: This compatibility research is important to provide practical recommendations in hospitals. As the best of our knowledge, the drugs used in the study have never been tested for compatibility, so they provide novelty value. In addition, this study perhaps will provide positive enthusiasm for research in the herbal field, because herbal medicines can be made in injection preparations that are stable and compatible with other injection preparations.

This research has used an excellent method to test the compatibility of injectable drugs. The manuscript also shows the method in detail. Improvement may be required regarding this

1. How is the final justification for incompatibility taking into account all tests. So that in the results section, readers can understand where the number 27 incompatible drugs comes from?

It is necessary to state that "the incompatibility results of one of the tests conclude that there is incompatibility". Or it would be clearer if the justification were shown in the picture. As an illustration, perhaps you can see figure 1 from this publication Compatibility of acetaminophen with central nervous system medications during simulated Y-site injection. Anaesthesiology Intensive Therapy. 2020;52(1):23-27. doi:10.5114/ait.2020.92684.

2. The results section will be easier to read if it is displayed in a table with columns of results per test.

3. In the discussion section it may be necessary to briefly discuss how to manage incompatible drugs for practical purposes. Please refer to the results of this publication "Management of Y-Site Incompatibility of Intravenous Medication: A Scoping Review." Indonesian Journal of Pharmacy/Indonesian Pharmacy Magazine 33.3 (2022).

Reviewer #2: Overall, the manuscript presents interesting findings, but the data is challenging to interpret. The data needs to be in a reader-friendly format. Easy to review, compare, and reference.

Page 4-5. 2.4 . Tyndall Beam Assessment Lines 76-78

Grammatical errors – Please revise - the sentence is confusing as written.

Page 5. 2.5. Turbidity Line 82

Grammatical errors – Please revise - the sentence is confusing as written.

Page 5. 2.6. Particle Line 89

Grammatical errors – Please revise - the sentence is confusing as written. When you say “three times” it is unclear what you are describing. It is an incomplete thought, are you saying you tested the same sample three times or you tested three samples?

Grammar – mixing present and past tense

Page 10 - Lines 181-182. A confusing explanation for the change in pH.

A 10% change in the pH of a solution of 4.0 is 0.4 and for 10.0 is 1.0. The authors need to clarify their statements regarding pH measurements and the 10% cut-off.

“When the pH of a solution change is 0.4 to 1.0 it is not 10% unless you are specifically referencing a change in pH of 4.0 to 4.4.

Tables are needed to understand the results for the different admixtures. Readers appreciate tables conveying the key findings.

6. PLOS authors have the option to publish the peer review history of their article (what does this mean?). If published, this will include your full peer review and any attached files.

Reviewer #1: **Yes: **Suci Hanifah

Reviewer #2: No

---

## [Author Response · Author response to Decision Letter 0]

22 Jan 2024

Dear Editor and Reviewers,

Thanks very much for taking your time to review this manuscript. I really appreciate all your comments and suggestions! Please find my itemized responses in below and my revisions/corrections in the re-submitted files. 

Thanks again! 

Reviewers’ Comments: 

Reviewer #1: This compatibility research is important to provide practical recommendations in hospitals. As the best of our knowledge, the drugs used in the study have never been tested for compatibility, so they provide novelty value. In addition, this study perhaps will provide positive enthusiasm for research in the herbal field, because herbal medicines can be made in injection preparations that are stable and compatible with other injection preparations.

This research has used an excellent method to test the compatibility of injectable drugs. The manuscript also shows the method in detail. Improvement may be required regarding this.

Comments: 

1.How is the final justification for incompatibility taking into account all tests. So that in the results section, readers can understand where the number 27 incompatible drugs comes from?

It is necessary to state that "the incompatibility results of one of the tests conclude that there is incompatibility". Or it would be clearer if the justification were shown in the picture. As an illustration, perhaps you can see figure 1 from this publication Compatibility of acetaminophen with central nervous system medications during simulated Y-site injection. Anaesthesiology Intensive Therapy. 2020; 52(1): 23-27. doi:10.5114/ait.2020.92684.

Response

Thank you for valuable review and comments on our manuscript. You are right. We have combined these justification of physical incompatibilities into a figure from the publication you suggested us.

2. The results section will be easier to read if it is displayed in a table with columns of results per test.

Response

Thank you for your professional suggestion. We have added all tables under results section per test and made some simplifications according to your valuable comments.

3. In the discussion section it may be necessary to briefly discuss how to manage incompatible drugs for practical purposes. Please refer to the results of this publication "Management of Y-Site Incompatibility of Intravenous Medication: A Scoping Review." Indonesian Journal of Pharmacy/Indonesian Pharmacy Magazine 33.3 (2022).

Response

Thank you for valuable review and professional suggestion to our manuscript. We have added discussion on how to manage incompatible drugs for practical purposes in the discussion section. We also carefully studied two literatures that you recommended to us at the same time.

Reviewer #2: Overall, the manuscript presents interesting findings, but the data is challenging to interpret. The data needs to be in a reader-friendly format. Easy to review, compare, and reference.

Comments: 

1.Page 4-5. 2.4 . Tyndall Beam Assessment Lines 76-78

Grammatical errors – Please revise - the sentence is confusing as written.

Page 5. 2.5. Turbidity Line 82

Grammatical errors – Please revise - the sentence is confusing as written.

Page 5. 2.6. Particle Line 89

Grammatical errors – Please revise - the sentence is confusing as written. When you say “three times” it is unclear what you are describing. It is an incomplete thought, are you saying you tested the same sample three times or you tested three samples?

Grammar – mixing present and past tense

Response

Thank you for your suggestion. We have refined and modified the entire grammar and sentences in detail.

2.Page 10 - Lines 181-182. A confusing explanation for the change in pH. A 10% change in the pH of a solution of 4.0 is 0.4 and for 10.0 is 1.0. The authors need to clarify their statements regarding pH measurements and the 10% cut-off. “When the pH of a solution change is 0.4 to 1.0 it is not 10% unless you are specifically referencing a change in pH of 4.0 to 4.4.

Response

We are grateful for the suggestion. To be more clear and in accordance with your concerns, we have modified the description in manuscript as follows: “all solutions were within the range of 4.0 to 10.0. Compared with 0h, if the pH value of the solution changes by 10%, the pH result will change within 0.4~1.0 accordingly”

3.Tables are needed to understand the results for the different admixtures. Readers appreciate tables conveying the key findings.

Response

Thank you for your valuable comments on our paper. Your suggestions are very good. We have considered this issue before, but because the simplified table may damage the integrity of this article, we did not make major adjustments. However, we have made some simplifications according to your valuable comments, hoping to make it easier for readers to understand. I am very eager for my manuscript to be accepted by your journal.

---

## [Editor Report · Decision Letter 1]

15 Feb 2024

Physical compatibility of Xuebijing injection with 53 intravenous drugs during simulated Y-site administration

PONE-D-23-34697R1

Dear Dr. Liu,

We’re pleased to inform you that your manuscript has been judged scientifically suitable for publication and will be formally accepted for publication once it meets all outstanding technical requirements.

Kind regards,

Adeel Sattar, Ph.D

Academic Editor

PLOS ONE

Additional Editor Comments (optional):

I think authors improved this article according to the suggestions raised by reviewers. I would like to accept this manuscript for publication
---

## [Editor Report · Acceptance letter]

13 Mar 2024

PONE-D-23-34697R1 

PLOS ONE

Dear Dr. Liu, 

I'm pleased to inform you that your manuscript has been deemed suitable for publication in PLOS ONE. Congratulations! Your manuscript is now being handed over to our production team.

Kind regards, 

on behalf of

Dr. Adeel Sattar 

Academic Editor

PLOS ONE